# Biomechanical Load of Neck and Lumbar Joints in Open-Surgery Training

**DOI:** 10.3390/s23156974

**Published:** 2023-08-05

**Authors:** Ce Zhang, Charlotte Christina Roossien, Gijsbertus Jacob Verkerke, Han Houdijk, Juha M. Hijmans, Christian Greve

**Affiliations:** 1Department of Rehabilitation Medicine, University of Groningen, University Medical Center Groningen, Hanzeplein 1, 9713 GZ Groningen, The Netherlands; g.j.verkerke@umcg.nl (G.J.V.); j.m.hijmans@umcg.nl (J.M.H.); c.greve@umcg.nl (C.G.); 2Center for Human Movement Sciences, University of Groningen, University Medical Center Groningen, Hanzeplein 1, 9713 GZ Groningen, The Netherlands; c.c.roossien@umcg.nl (C.C.R.); h.houdijk@umcg.nl (H.H.); 3Department of Biomechanical Engineering, University of Twente, Drienerlolaan 5, 7522 NB Enschede, The Netherlands

**Keywords:** inertial measurement unit (IMU), open surgery, ergonomic assessment, joint moment, joint compression force

## Abstract

The prevalence of musculoskeletal symptoms (MSS) like neck and back pain is high among open-surgery surgeons. Prolonged working in the same posture and unfavourable postures are biomechanical risk factors for developing MSS. Ergonomic devices such as exoskeletons are possible solutions that can reduce muscle and joint load. To design effective exoskeletons for surgeons, one needs to quantify which neck and trunk postures are seen and how much support during actual surgery is required. Hence, this study aimed to establish the biomechanical profile of neck and trunk postures and neck and lumbar joint loads during open surgery (training). Eight surgical trainees volunteered to participate in this research. Neck and trunk segment orientations were recorded using an inertial measurement unit (IMU) system during open surgery (training). Neck and lumbar joint kinematics, joint moments and compression forces were computed using OpenSim modelling software and a musculoskeletal model. Histograms were used to illustrate the joint angle and load distribution of the neck and lumbar joints over time. During open surgery, the neck flexion angle was 71.6% of the total duration in the range of 10~40 degrees, and lumbar flexion was 68.9% of the duration in the range of 10~30 degrees. The normalized neck and lumbar flexion moments were 53.8% and 35.5% of the time in the range of 0.04~0.06 Nm/kg and 0.4~0.6 Nm/kg, respectively. Furthermore, the neck and lumbar compression forces were 32.9% and 38.2% of the time in the range of 2.0~2.5 N/kg and 15~20 N/kg, respectively. In contrast to exoskeletons used for heavy lifting tasks, exoskeletons designed for surgeons exhibit lower support torque requirements while additional degrees of freedom (DOF) are needed to accommodate combinations of neck and trunk postures.

## 1. Introduction

Surgeons are at high risk of developing musculoskeletal symptoms (MSS) such as neck and back pain [1]. The high prevalence of MSS among surgeons can lead to sick leave, reduced productivity and reduced surgical quality [2]. MSS may require medical treatment, reduce the career length of surgeons, and affect the safety of patients. The incidence of MSS also significantly rises with age, i.e., as ageing causes muscle loss. Consequently, the burden of musculoskeletal conditions is expected to increase [3,4,5,6]. Compared with laparoscopic surgery, open surgery demands larger neck and trunk flexion angles, which increases spinal loading and the risk for developing MSS [7]. Among open-surgery surgeons, the most prevalent MSS regions are the neck (80.7%) and lower back (65.3%) [8]. Especially prolonged working in the same posture [9], unfavourable working postures (e.g., flexion with lateral bending and rotation of the neck and lower back) [9,10,11] are thought to be the most important biomechanical risk factors for developing MSS. Exoskeletons might be potential solutions to reduce the load on spinal structures during surgery by providing supportive forces and reducing spinal muscle activity and joint reaction forces [12,13].

Currently, available exoskeletons are mainly designed for reducing peak loads while lifting heavy objects [14,15,16]. However, while peak spinal joint loads and compression forces during open surgery are relatively low compared to lifting tasks, spinal structures are loaded for longer durations. Prolonged loading is often accompanied by unfavourable spinal postures including flexion and rotation increasing tissue strain [17]. In addition, while the neck can be kept straight during heavy lifting tasks, during surgery the orientation is mainly flexed. A detailed understanding of the spine biomechanics during open surgery is important for developing effective preventive devices for MSS in surgeons and to determine (1) which spinal postures of surgeons are most prevalent, (2) how much external support is required to unload spinal structures at the neck and back, and (3) which degrees of freedom should be available to prevent interference with the surgical procedure.

Previous studies have evaluated the kinematics of surgeons during surgery tasks [18,19,20,21,22,23], but a detailed analysis of joint angle distribution and joint loads during surgery has not been established yet. To address this need, a validated inertial measurement unit (IMU) system is used to obtain the working postures of surgical trainees [24]. In addition, a validated musculoskeletal model is used to establish the neck and lumbar joint loads during open-surgery training [25]. The objective of this study is to establish the biomechanical profile of the neck and lumbar joint of surgical trainees, estimate the joint load of the lower back and neck during open surgery (training), and provide recommendations (e.g., required joint range of motion and support moment/force) for the design of ergonomic devices to reduce the risk of MSS.

## 2. Materials and Methods

### 2.1. Participants

In this study, we included eight healthy surgical trainees (3 males, 5 females; age: 31 ± 3 years; height: 170.0 ± 7.7 cm; body mass: 74.1 ± 11.6 kg; body mass index: 23.5 ± 2.2 kg/m^2^; surgical training experience: 2 ± 1 years). The Medical Ethics Review Board of the University Medical Center Groningen (UMCG) approved the study (METc 2022/385). Prior to the study, all participants were provided with information regarding the aim of the research and subsequently signed written informed consent. The inclusion criteria were as follows: (1) being at least 18 years old, and (2) have prior experience in open-surgery training as a surgical trainee. Participants were excluded if they had a history of MSS or other self-reported orthopaedic or neuromuscular complaints that could impact their movement patterns.

### 2.2. Experimental Set-Up

#### 2.2.1. IMU Sensors

Five Xsens MTw inertial measurement unit sensors (Xsens Technologies B.V., Enschede, The Netherlands) were placed on the back of the head, the proximal sternum, the upper spine at the level of spinous process of T5 and above T10, and on the back of the pelvis (Figure 1) using elastic bands and tape. The IMU’s positive *z*-axis pointed forward, and the positive *y*-axis pointed up [24]. The Awinda Station (Xsens Technologies B.V., Enschede, The Netherlands) and MT manager software (v2021.0.1 build 6752) was used for synchronizing and collecting data [26]. Acceleration and orientation of the sensors were recorded at 100 Hz.

#### 2.2.2. Experimental Procedures

The data collection took place in the surgical training room in the Skills Center of the University Medical Center Groningen. The training room was used to simulate a real operating room environment, complete with surgical tables, surgical lights, and other surgical equipment. The surgical training focused on open abdominal surgery performed on cadavers, including appendectomy, fundoplication, and inguinal herniorrhaphy. According to informal interviews with surgical instructors, the movements of surgical trainees during their training were considered to be comparable to those performed during real surgery. Before the surgical training, the IMUs were placed on the participant and were calibrated with the participant standing in an erect posture. The following anthropometric information was gathered: head length (antero–posterior dimension of the head, measured from the most anterior point and the most posterior of the head), width (transverse dimension of the head, maximum width of the head) and height (vertical dimension of the head, measured from top of the head to the C7), pelvis width (distance between left and right anterior superior iliac spine, distance between left and right posterior superior iliac spine) and length (distance between the anterior superior iliac spine and posterior superior iliac spine), the distance between C7 and S1, and the distance between the left and right acromioclavicular joints. After completing the above steps, the surgical trainee put on a surgical gown and started performing open abdominal surgery (shown in Figure 2). The measurement duration of open surgery (training) lasted the whole duration of the surgery and ranged from 15 min to 88 min (average 50 min).

#### 2.2.3. Model

In this study, a customized OpenSim musculoskeletal model was used for calculating neck and trunk kinematics and estimating joint moments and reaction forces (shown in Figure 3). The customized model was adapted from a validated full-body thoracolumbar spine model (Fullbody_OS4.x_v2.0) [25], with the exclusion of the lower extremity muscles and internal and external intercostals to improve computational efficiency. The lumbar angle was defined as the angle between the trunk and pelvis segment at the L5 and S1 level. The neck angle was defined as the angle between the head and trunk segment, at the C7 and T1 vertebrae level. The customized musculoskeletal model constrained all the degrees of freedom except for pelvis tilt, pelvis obliquity, pelvis rotation, pelvis translation_x (anteroposterior axis), pelvis translation_y (longitudinal axis), pelvis translation_z (transverse axis), lumbar flexion/extension (lumbar FE), lumbar lateral bending (lumbar LB), lumbar angular rotation (lumbar AR), neck flexion/extension (neck FE), neck lateral bending (neck LB), and neck angular rotation (neck AR).

### 2.3. Data Analysis

#### 2.3.1. Kinematics Calculation

The Xsens fusion filter algorithm [27] was utilized to export 3-axis acceleration and direction cosine matrix data with the MT manager software (v2021.0.1 build 6752) [26]. Gap filling was carried out when less than 10 consecutive data points were lost. The collected anthropometric information was used for obtaining the scaled model for each participant. Customized MATLAB 2020b (MathWorks, Natick, MA, USA) scripts were used to perform the standard OpenSim 4.3 (simtk.org/projects/opensim) IMU inverse kinematic workflows for IMU with the scaled model and orientation files. The OpenSim IMU inverse kinematic algorithm iterates through each time step (frame) of motion and computes generalized coordinate values to position the model in a pose that closely aligns with the experimental IMU orientations. This alignment process is mathematically expressed as a weighted least squares problem in Equation (1) [28,29].
(1)$$\min_{q}\underset{i\in IMUs}{\sum \omega_{i}\theta_{i}^{2}}$$
where *q* is the vector of generalized coordinates, *ω_i_* is the weight corresponding to the orientation of *IMU_i_*, and *θ_i_* is the angle component of the orientation error expressed by an axis-angle representation.

The neutral position is defined as 0 degrees for neck and lumbar joints. The mean angle and mean range of motion (ROM) with the standard deviation of neck and lumbar joints (FE, LB and AR) across eight participants were computed. To quantify the frequency of neck and trunk postures during surgery, the average durations of the selected neck and lumbar joint angle configurations for each 10-degree range normalized to total surgery duration were displayed in histograms. The negative angles indicate flexion, left lateral bending and clockwise rotation angle.

#### 2.3.2. Joint Moment and Reaction Force Calculation

The standard OpenSim Inverse Dynamic workflows were used to calculate the net moment of the neck and lumbar joints based on the filtered kinematic data (6 Hz lowpass) and scaled model for each participant.

Based on the kinematic results, the model’s motion, including its generalized positions, velocities, and accelerations, is already known. The OpenSim inverse dynamics algorithm uses the known motion (kinematics results) of the model to compute the unknown generalized forces *τ* by solving Equation (2) [30].
(2)$$\tau =M(q)\overset{¨}{q}+C(q,\overset{¨}{q})+G(q)$$
where $\tau \in R^{N}$ is the vector of generalized forces; $q,\overset{˙}{q},\overset{¨}{q}\in R^{N}$ are the vectors of generalized positions, velocities, and accelerations, respectively; $M(q)\in R^{N\times N}$ is the musculoskeletal system mass matrix; $C(q,\overset{˙}{q})\in R^{N}$ is the vector of Coriolis and centrifugal forces; $G(q)\in R^{N}$ is the vector of gravitational forces; and *N* is the number of degrees of freedom.

The participants’ average duration was 50 min, and their neck and trunk postures were mainly static. To reduce computational burden while maintaining meaningful analyses, the OpenSim Static Optimization (SO) function was employed to compute neck and trunk muscle forces at 10 s intervals. The static optimization algorithm in OpenSim utilizes the model’s known motion to solve the equations of motion for the unknown generalized forces and moments, such as joint torques. This solution is subject to specific muscle activation-to-force conditions as described in Equations (3) and (4) [30]:(3)$$\sum_{m=1}^{n}\left[a_{m}f(F_{m}^{0},l_{m},v_{m})\right]r_{m,j}=\tau_{j}$$
while minimizing the objective function (muscle activations):(4)$$J=\sum_{m=1}^{n}\left(a_{m}\right)p$$
where *n* is the number of muscles in the model; *a_m_* is the activation level of muscle *m* at *a* discrete time step; $F_{m}^{0}$ is its maximum isometric force; *l_m_* is its length; *v_m_* is its shortening velocity; $f(F_{m}^{0},l_{m},v_{m})$ is its force-length-velocity surface; *r_m,j_* is its moment arm about the *j*th joint axis; *τ_j_* is the generalized force acting about the *j*th joint axis; and *p* is a user defined constant.

Then, joint reaction (JR) analysis was used to obtain the neck and lumbar joint compression forces. Muscle tendon actuators were assigned at each degree of freedom in the model to balance the kinetic requirements for SO and JR analysis [31]. Neck and lumbar joint reaction forces (compression force) were expressed in the parent body reference frame, the parent body frame for the neck joint was the T1 segment, and the parent body frame for the lumbar joint was the pelvis segment.

Joint moments and reaction forces were normalized by body weight. Histograms of the normalized neck and lumbar joint moment and joint reaction force across eight participants were used to show the distribution of neck and lumbar joint load. A histogram bin of 0.01 Nm/kg for neck flexion and lateral bending moment, 0.005 Nm/kg for neck rotation moment, 0.2 Nm/kg for lumbar flexion and lateral bending moment, and 0.01 Nm/kg for lumbar rotation moment were used. The positive moments in the neck and lumbar joints express flexion, left lateral bending, and clockwise rotation moment. Additionally, a histogram bin of 5 N/kg was used for lumbar compression force, whereas for neck compression force the bin was 0.5 N/kg.

## 3. Results

### 3.1. Kinematics

#### 3.1.1. Mean Working Posture and Range of Motion

Table 1 shows the mean and standard deviation of lumbar and neck joint angles across eight participants during open-surgery training. The surgeons typically maintained a flexed lumbar and neck posture during the open-surgery training, with mean flexion angles of approximately 20.4 ± 10.0 and 19.8 ± 6.69 degrees, respectively. The mean working posture angle of lumbar and neck lateral bending and rotation were close to 0 degrees with a standard deviation of 2.23~5.05 degrees.

The mean range of motion (ROM) of the lumbar joint was between 45 and 55 degrees in the sagittal, frontal and transverse planes. The neck joint had a greater ROM than the lumbar joint, with a mean flexion and extension ROM of 102 ± 25.6 degrees, lateral bending ROM of 68.6 ± 11.6 degrees, and rotation ROM of 149 ± 11.8 degrees.

#### 3.1.2. Joint Angle Distribution

Figure 4a shows neck and lumbar joint angle distribution. The horizontal axis represents the range of angles, the vertical axis represents the percentage of time, the vertical line represents standard deviation. In Figure 4a, it is shown that the neck flexion angle during open-surgery training was predominantly between 20 and 30 degrees, accounting for 32.8% of the duration (SD 15.6%). In the frontal and transversal planes, the neck joint angles were close to the neutral position, between −5 and 5 degrees, for 41.3% and 37.5% of the time, respectively (Figure 4b,c).

The lumbar joint was flexed 10~20 and 20~30 degrees during 41.8% (SD 19.4%) and 27.1% (SD 20.1%) of the surgery duration (Figure 4d). Lumbar lateral bending and rotation angles were distributed between −5 to 5 degrees for 48.3% and 57.9% of the time, respectively (Figure 4e,f).

The 2D histogram plots (Figure 5) illustrate the distribution of combined joint motions across joints and degrees of freedom. Combined lumbar and neck flexion angles varied largely between participants with the highest distributions in 10 to 30 degrees lumbar flexion and 10 to 40 degrees neck flexion. The lumbar flexion with rotation had a more concentrated range than neck combination postures. More combined movements are seen in the neck joint (Figure 5e) than in the lumbar joint (Figure 5c).

### 3.2. Kinetics

#### 3.2.1. Mean Lumbar and Neck Moment

The mean lumbar and neck moments are shown in Table 2. The surgeons endured a mean external lumbar and neck flexion moment of 0.503 and 0.0419 Nm/kg, respectively. The mean lumbar and neck lateral bending and rotation moments were all close to 0.

#### 3.2.2. Lumbar and Neck Moment Distribution

Figure 6a shows that the external neck flexion moment was mainly distributed around 0.04~0.06 Nm/kg (53.8% of the duration), while the neck lateral bending and rotation moments were centered around −0.005~0.005 Nm/kg (26.8% of the duration) and −0.0001~0.0001 Nm/kg (51.4% of the duration), respectively (Figure 6b,c). Figure 6d shows that the normalized lumbar flexion moment was between 0.4 and 0.6 Nm/kg for 35.5% of the duration of the surgery, lumbar lateral bending and rotation moments were centered around −0.1~0.1 Nm/kg (44.0% of the duration) and −0.05~0.05 Nm/kg for 56.0% of the duration, respectively (Figure 6e,f).

#### 3.2.3. Lumbar and Neck Joint Reaction Forces

The normalized lumbar and neck joint reaction force is shown in Table 3. The mean normalized lumbar joint compression forces were around 17.0 N/kg and the neck joint was subject to compression forces of 2.11 N/kg. The lumbar joint experienced a normalized maximum compression force of approximately 35.9 N/kg, and the neck joint maximum normalized compression force was 3.63 N/kg.

During open-surgery training, the lumbar compression force was between 15 and 20 N/kg for 38.2% (SD 19.0%) of the time. Meanwhile, the neck compression force fell within the range of 2.0~2.5 N/kg for 32.9% (SD 13.1%) of the duration (Figure 7).

## 4. Discussion

### 4.1. Biomechanical Profile during the Open-Surgery Training

During open-surgery training, surgeons tend to maintain a flexed lumbar and neck posture during the surgical procedure, which is consistent with findings from previous ergonomic studies [23,32]. Specifically, we observed an average neck angle of around 20 degrees. Previous studies reported an angle of around 40 degrees [23,32]. This discrepancy can however be attributed to differences in the definition of the neck angle. In our study, the neck angle was defined relative to the trunk, whereas in the previous studies, it was defined relative to the neutral position. We have conducted further analysis building upon previous research, focusing on the distribution of angles in both the lumbar and neck joints during the surgical procedures (Figure 4 and Figure 5). During open-surgery training, surgeons tend to maintain a flexed lumbar and neck posture for more than 70% of the surgical procedure (Figure 4a,d). It is worth noting that although the average values for neck and lumbar lateral bending and rotation are close to 0 degrees, there is still a noticeable distribution observed within the range of −15 to −5 and 5 to 15 degrees. Additionally, we investigated the combined postures of the neck and trunk (e.g., neck flexion with lateral bending and rotation, and trunk flexion with lateral and rotation). This distribution should be taken into consideration when implementing ergonomic interventions (e.g., exoskeletons). For example, it may be beneficial to add joints to an exoskeleton that allow for lumbar joint rotation and lateral bending movement [33]. Otherwise, the wearer may experience discomfort when performing activities that involve these combined movements, such as asymmetrical lifting tasks [34]. Regarding neck support, [35] and [36] propose additional degrees of freedom (DOF) for rotation during neck flexion, but they do not address the DOF for lateral bending, which may affect movements in the frontal plane for surgeons. Therefore, it is recommended to allow neck and trunk movements in the frontal and transverse planes in the design of exoskeletons/devices that provide support.

In addition to the kinematic analysis, our study quantified the moments at the neck and lumbar joints, as well as the compression on the lumbar and neck joints during the simulated surgical procedure. The results revealed that open-surgery training results in a maximum lumbar flexion moment ranging between 1.0 and 1.2 Nm/kg, with the majority of moments falling within 0.4~0.6 Nm/kg. Additionally, the maximum lumbar compression force during open-surgery training was found to be 35.9 N/kg. It should be noted that previous research [37] on lifting objects has indicated that the maximum lumbar flexion moment can be as high as 3 Nm/kg, while the maximum lumbar compression force during lifting activities can be as high as 63.5 N/kg. Lifting tasks often involve carrying higher loads over shorter durations, which necessitates larger peak joint moments. In contrast, surgical tasks frequently require sustained work over extended periods, and surgeons must maintain specific postures for long durations, resulting in different moment and force requirements on the lumbar joint [38]. These different tasks demand to impose distinct design constraints on support systems specific to surgery as outlined in the next section.

### 4.2. Implications for Designing a Neck and Trunk Exoskeleton

The biomechanical assessment of the spine during open surgery reveals valuable information for developing exoskeletons to reduce the risk of MSS. The surgeons mostly operated in a range of 10 to 30 degrees of lumbar flexion (68.9% of duration) and 10 to 40 degrees of neck flexion (71.6% of duration). An exoskeleton should provide support in the range of 10 to 30 degrees lumbar flexion with 0.4~0.6 Nm/kg support torque, and 10 to 40 degrees of neck flexion with 0.04~0.06 Nm/kg. Considering the variability in surgeons’ movements, designing exoskeletons that can be tuned to the individual is also important. The observed standard deviations in the histogram of joint angles and loads (Figure 4, Figure 5, Figure 6 and Figure 7) indicate that surgeons have diverse movement patterns and may require personalized support based on their specific movement and anatomical characteristics. By incorporating customization and adjustability features (such as stiffness or joint range of motion) into the exoskeleton design, it becomes possible to provide tailored support to individual surgeons and hence improve acceptance.

Compared with the exoskeletons used for lifting heavy objects, the required support torque for surgeon exoskeletons is relatively low, and the strength limit of the exoskeleton can be lower than the exoskeleton for lifting. Therefore, it should be possible to make these exoskeletons lighter and more compact. The ideal weight for an exoskeleton should be less than 3% of the total body weight [39]. For the neck exoskeleton, it is worth noting that the required support torque for fully supporting the neck of 0.04~0.06 Nm/kg can be easily achieved. On the other hand, if the support torque is too high, the surgeon will need to actively contract their abdominal and neck flexor muscles to overcome the resistance of the exoskeleton. This would provide support for surgeons without compromising their ability to maintain precision and move without restriction during surgery.

In summary, based on this study we recommend that trunk exoskeletons should provide support mainly on 10~30 degrees of lumbar flexion with 0.4~0.6 Nm/kg and allow 50 degrees in trunk flexion and 10 degrees in trunk extension, 25 degrees in left and right lateral bending and rotation. An effective neck exoskeleton should primarily provide flexion support within the range of 10~40 degrees, and a support torque of 0.04~0.06 Nm/kg. It is also recommended to allow for lateral bending and rotation of up to 25 degrees and 50 degrees, respectively. Finally, the exoskeleton should be designed taking the type of surgery into account, in this study we focus on open surgery, postures and workload can be different for laparoscopic surgery [7].

### 4.3. Limitation and Recommendation

There were several limitations in this study. Firstly, the reaction force on the hands was not considered when calculating the moment and reaction forces of the lumbar joint. Additionally, the external moments of the arms were not considered in the IMU method, leading to an underestimation of the lumbar moment and compression force. To address this limitation, a validation study (See Appendix A for detail) was conducted to assess the effect of arm movement on the lumbar joint biomechanical profile. Two scenarios were examined: trunk flexion with the arm stretched (Figure A3) and standing straight with the arm fully stretched (Figure A4). Trunk flexion with the arm stretched was used to simulate surgery postures where arm movements are typically involved. In this scenario, the lumbar joint moments and compression forces were underestimated by 15.2% and 11.8%, respectively. The scenario of standing straight with the arm fully stretched aimed to simulate the worst-case scenario where arm movements were disregarded. The results showed that the estimations in this scenario led to a maximum underestimation of 23.8% for moments and 21.1% for compression forces. Therefore, for the design of exoskeletons, it is crucial to increase the moment requirement by at least 15.2% when aiming at full support. Furthermore, it is recommended that future studies estimating the neck and lumbar joint biomechanical profiles during open surgery should account for the contribution of arm segments during kinetic analyses. In the current study there are large SDs in the postures and loads, a larger sample size would provide more accurate data on the range of inter-individual differences in biomechanical profiles, which can be used to design customizable exoskeletons.

## 5. Conclusions

In our study, we established the biomechanical profile of the neck and lumbar joint of surgical trainees during open surgery (training) using IMU and musculoskeletal modelling. The movements of surgeons involve complex combinations of neck and trunk postures, and the biomechanical load associated with these movements is generally lower compared to the task of lifting heavy objects. Therefore, exoskeletons aiming to reduce biomechanical loading of the lumbar spine should provide adjustable support forces for neck and lumbar joint flexion while allowing axial rotation and lateral bending motions.

## Figures and Tables

**Figure 1 sensors-23-06974-f001:**
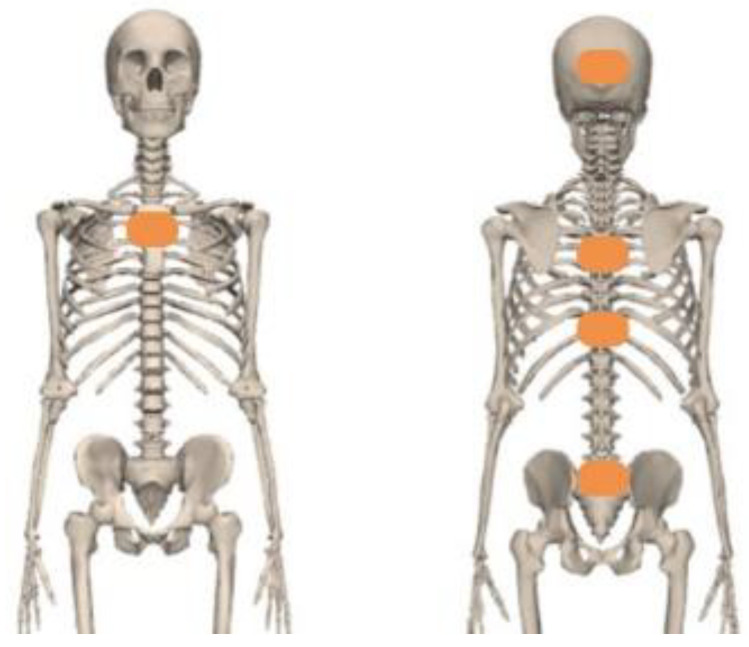
IMU sensors (orange rectangles) placement.

**Figure 2 sensors-23-06974-f002:**
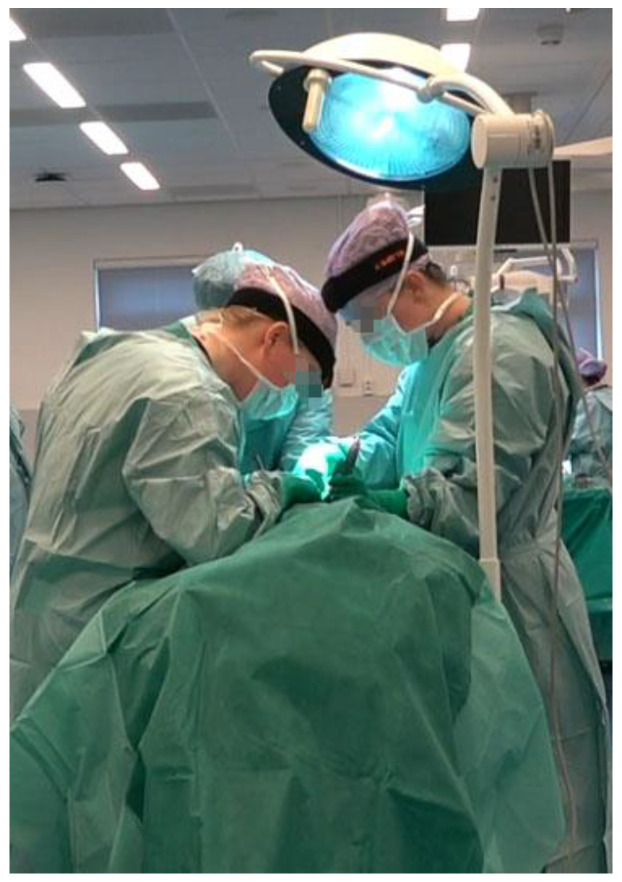
Example of surgery training task.

**Figure 3 sensors-23-06974-f003:**
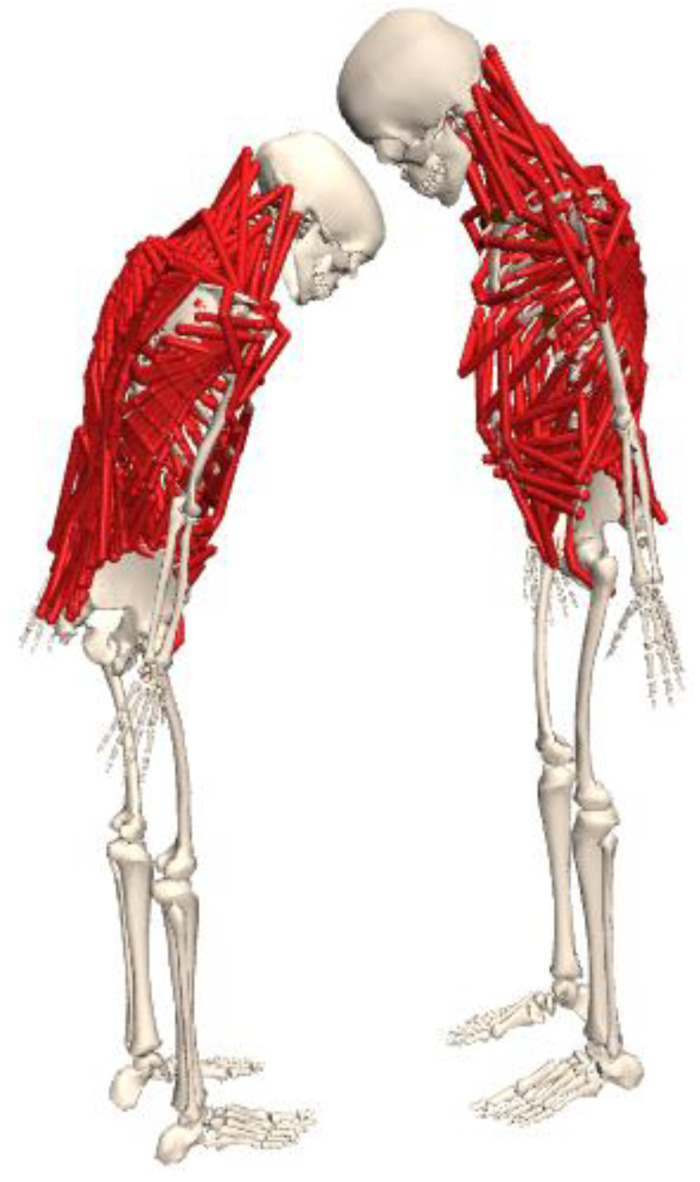
Musculoskeletal model.

**Figure 4 sensors-23-06974-f004:**
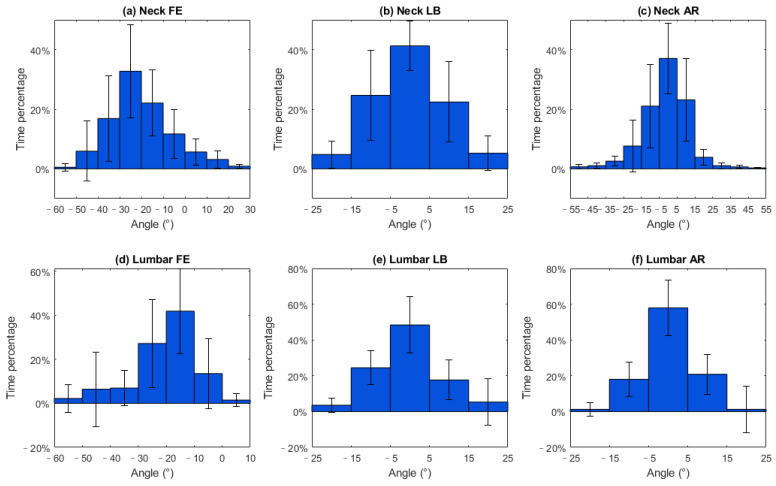
Histograms of the angles of the neck and lumbar joints during open-surgery training. Footnote: The horizontal axis represents the angle and the vertical axis represents the time percentage. The vertical lines on the graph represent the standard deviation.

**Figure 5 sensors-23-06974-f005:**
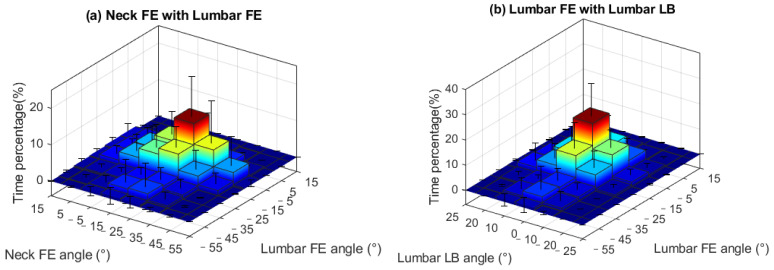
Combination of neck and trunk postures during the open-surgery training. Footnote: The horizontal axes represent the angle and the vertical axis represents the time percentage. The vertical lines on the graph represent the standard deviation.

**Figure 6 sensors-23-06974-f006:**
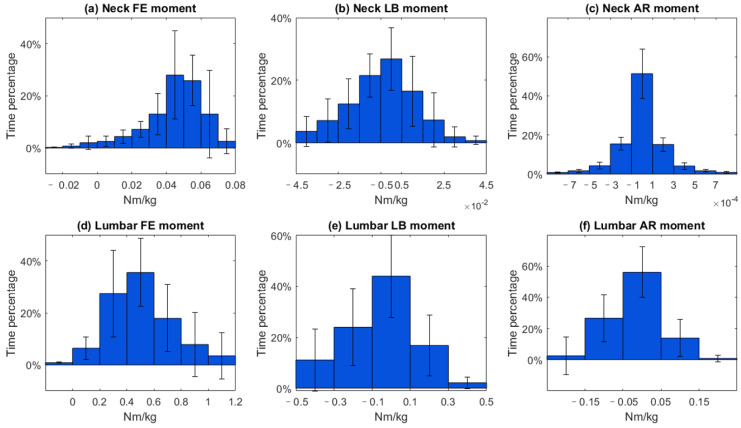
Histogram of normalized moment for neck and lumbar joint during open-surgery training. Footnote: The horizontal axis represents the normalized moment and the vertical axis represents time percentage. The vertical lines on the graph represent the standard deviation.

**Figure 7 sensors-23-06974-f007:**
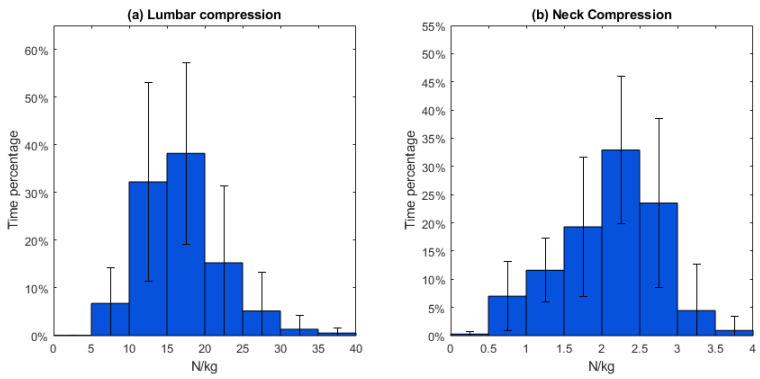
Lumbar and neck joint reaction forces distribution during open-surgery training. Footnote: The horizontal axis represents normalized compression force and the vertical axis represents the time percentage. The vertical lines on the graph represent the standard deviation.

**Table 1 sensors-23-06974-t001:** Mean working postures and range of motion (n = 8).

	Lumbar FE	Lumbar LB	Lumbar AR	Neck FE	Neck LB	Neck AR
Mean posture (°)	−20.4	−0.373	0.291	−19.8	0.0393	−1.65
SD (°)	10.0	5.05	2.23	6.69	5.04	4.81
ROM (°)	53.4	45.6	54.1	102	68.6	149
SD (°)	12.2	10.4	17.4	25.6	11.6	11.8

FE: flexion/extension, LB: lateral bending, AR: angular rotation, and ROM: range of motion.

**Table 2 sensors-23-06974-t002:** Mean normalized lumbar and neck moment (n = 8).

Moment/Mass (Nm/kg)	Lumbar FE	Lumbar LB	Lumbar AR	Neck FE	Neck LB	Neck AR
Mean	0.503	0.0101	0.00210	0.0419	4.65 × 10^−4^	−3.24 × 10^−9^
SD	0.179	0.0907	0.0282	0.00743	0.00842	1.38 × 10^−8^

**Table 3 sensors-23-06974-t003:** Mean normalized lumbar and neck joint reaction force (n = 8).

N/kg	Lumbar Compression	Neck Compression
Mean	17.0	2.11
SD	3.71	0.284
Max.	35.9	3.63
SD	17.7	0.913

## Data Availability

The data presented in this study are available on request from the corresponding author. The data are not publicly available due to ethical and privacy reasons.

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
