# Peer review of "Biomechanical Load of Neck and Lumbar Joints in Open-Surgery Training"

_sensors, 2023, doi:10.3390/s23156974_

Round 1

Reviewer 1 Report

Dear Authors,

the manuscript I am reviewing is very interesting, with high scientific potential. However, some changes are needed in the presentation of the results and the style of discussion in order to improve the manuscript.

Attached are my comments on the reviewed article.

Reviewer 2 Report

In the study, the authors established the biomechanical profile of the neck and lumbar joint of surgical trainees during open surgery (training) using IMU and musculoskeletal modelling.The elaboration process of the paper involves several typical algorithms and modeling methods, such as the Xsens fusion filter algorithm  and a customized OpenSim musculoskeletal model . However, the overall content of the paper is closer to a test report rather than an academic paper, and the paper lacks core technological innovation points. It is recommended that the author make further supplements in this regard, otherwise it cannot meet the basic requirements for publication.

The English expression of this paper is relatively standardized and fluent.

Reviewer 3 Report

The manuscript investigates the joint angles and loads affecting the neck and lower back during open surgery. The angles of the neck and trunk were measured using IMU sensors, and the joint moments and compression forces were estimated using the OpenSim musculoskeletal model. The results will serve as a design guideline for exoskeletons aimed at reducing the risk of musculoskeletal symptoms for surgeons. This research is crucial for defining the problems associated with surgeon's exoskeletons, which should provide adjustable supporting forces while allowing for rotation and bending during surgical motions.

Only for the completeness, I would recommend keeping up brevity in overall manuscript for readers understanding.

1)       The authors measured only the angle and posture of neck and trunk using IMU sensor. The loading and compression were calculated by OpenSim musculoskeletal model. Then, the model accuracy was essential prerequisite for the biomechanical analysis. So, the relative standard deviation for the measured data and estimated one could help the verification.

2)       In page 5, 148th line, muscle force was computed at 10-second intervals. It is important to mention the rationale behind this choice of interval. Additionally, it may be beneficial to integrate the forces over the interval rather than compute it at discrete time points.

3)       In page 6, Figure 4. increase the font size of title for better visibility..

Round 2

Reviewer 1 Report

Dear Authors,

I have only cosmetic comments on the manuscript this round, which are listed below.

L42: Change „…expected to increase. [3–6].” to „..expected to increase [3–6].” (remove the full stop between the end of the sentence and the reference to literature).

L184: Move "Table 1. Mean working postures and range of motion (n=8). " to page 6, directly above the table.

L198-203: Move the text below Figure 4 directly above Figure 5. End the sentence (L203) with a full stop.

L390-391: Place the GS and RSD legends on one line below the table.

L394: Move the description of Figure A2 to page 13, directly under the figure.

L460 and L491: Change ";" to "." at the end of the reference.

Congratulations

Reviewer 2 Report

The corresponding formulas for certain core algorithms are still missing in the paper (see the previous review comments for details). It is recommended that the authors supplement relevant content and cite references.

The English writing ability of the paper meets the publication requirements.
